# Implementation of a Hybrid Educational Program between the Model of Personal and Social Responsibility (TPSR) and the Teaching Games for Understanding (TGfU) in Physical Education and Its Effects on Health: An Approach Based on Mixed Methods

**DOI:** 10.3390/children8070573

**Published:** 2021-07-02

**Authors:** Gregorio García-Castejón, Oleguer Camerino, Marta Castañer, David Manzano-Sánchez, José Francisco Jiménez-Parra, Alfonso Valero-Valenzuela

**Affiliations:** 1Department of Physical Activity and Sport, Faculty of Sport Sciences, University of Murcia, Santiago de la, Ribera, 30720 Murcia, Spain; gregodarkshoes@gmail.com (G.G.-C.); josefrancisco.jimenezp@um.es (J.F.J.-P.); avalero@um.es (A.V.-V.); 2Biomedical Research Institute of Lleida (IRBLLeida), 25192 Lleida, Spain; ocamerino@inefc.udl.cat (O.C.); castaner@inefc.udl.cat (M.C.); 3National Institute of Physical Education of Catalonia (INEFC), University of Lleida (UdL), 25192 Lleida, Spain

**Keywords:** mixed methodology, innovation, pedagogical models, teaching, secondary education

## Abstract

The present study investigates the effect of an educational program hybridized between the Model of Personal and Social Responsibility (TPSR) and the Teaching Games for Understanding (TGfU) in physical education classes on the health and psychosocial variables of students, as well as knowing the advantages and disadvantages of its implementation by teachers. The applied program lasted 11 weeks in two Secondary Education centers with a total of four teachers (two in the experimental group and two in the control group) and 99 students (55 from the control group and 44 from the experimental group). We use research methodology Mixed Methods with a quasi-experimental design, where students completed a questionnaire before and after the educational program and teachers were interviewed at the end of the intervention. The results of the questionnaires indicate significant improvements in the experimental group over time in terms of the intention to be physically active, as well as in autonomous motivation, the self-determination index, the index of psychological mediators, personal and social responsibility, and enjoyment. Moreover, the interviews show positive opinions regarding the organizational capacity of the session using this methodology and the interest of teachers in continuing to apply it in the future, as well as the need for initial and ongoing training for proper implementation. In conclusion, the hybridization between the TPSR and TGfU model is presented as an effective alternative to be applied in the educational context with the aim of improving young peoples’ intention to be physically active and psychological variables, such as motivation, responsibility, and enjoyment, in physical education classes.

## 1. Introduction

The World Health Organization (WHO) establishes that health is “a complete state of physical, mental and social well-being and not simply the absence of disease or illness” [1]. In this sense, the prevalence of physical inactivity and sedentary lifestyles are key factors that increase obesity in the worsening of general health [2]. For this situation, the WHO [3] establishes among its measures for people to be more physically active, the need for national policies to provide primary and secondary education centers with the necessary resources to develop a quality physical education (PE) that maintains physically active behaviors in children and adolescents throughout life.

PE is a useful subject to promote greater adherence to sports practice in students, and thus, achieve a better quality of life, health, motor skills, academic performance, or educational values [4]. The necessity to diversify teaching to meet the students’ needs and changes in society, has caused an evolution in the way of teaching [5], starting with teaching styles [6], and instructional models [7], up to model-based practices [8], and current pedagogical models [9]. Teaching based on pedagogical models consists of a methodological and pedagogical approach that establishes a relationship of interdependence among four key components within the teaching-learning process, such as the student, teacher, content, and context [10]. Moreover, it incorporates teaching styles into its structure to align learning outcomes with the needs of schoolchildren [11]. One of the most current pedagogical models in the field of PE to initiate young people in the practice of sport and promote healthy lifestyle habits is the one called Teaching Games for Understanding (TGfU) [9]. This model offers the possibility of teaching the basic principles of any sport through modified games [12], allowing students to learn and understand the framework, tactics, and technical skills of a sport discipline [13,14]. For this, PE teachers must consider the six phases into which the TGfU is divided can apply it in the classroom [13].

TGfU has a great impact on cognitive learning, pursuing to train competent students, capable of making decisions and solving tactical problems [13,14]. Contreras et al. [15] affirm that using this methodology actively favors the teaching and motivation of the students towards learning [16], as well as increasing the practice time of moderate and vigorous physical activity [17]. For Cocca et al. [12], some of these factors make TGfU one of the main models that PE teachers use to improve the student’s health. Another of the models that have stood out in recent years has been the Model of Personal and Social Responsibility (TPSR) [18], considered as an adequate pedagogical approach to promote education in values through the promotion of responsibility, autonomy, and the integral development of learners in their social environment [19]. Its implementation in the educational context is carried out progressively through the five levels of responsibility proposed by Hellison [18]: (1) Respect for the rights and feelings of others, (2) participation and effort, (3) personal autonomy, (4) helping others and leadership, and (5) transfer outside the classroom; that define behaviors and attitudes that enhance the capacities of students [20], and thus, acquire guidelines of individual and group responsibility in line with the implicit values of society [5]. Empirical research on TPSR has increased exponentially in recent years, where it has been applied in education and different areas confirming great adaptability to any context [21]. Moreover, it also reports benefits in health education [22], with a positive correlation between the responsibility and time spent practicing physical and sporting activity [23], and basic psychological needs, motivation, sportsmanship, and the intention to be physically active [24].

However, despite the multiple benefits that pedagogical models have provided [20,25,26,27], one that serves to cover all educational contents or contexts has not yet been found [5,7,28]. This limitation is due to the necessity to combine several significant parts or elements of these models [9], giving rise to the term known as hybridization of pedagogical models [28]. Fernández-Río et al. [5] show the existence of two essential pillars that build the foundational basis of this idea: (1) Situated Learning: Based on the connection between students, content, knowledge, and the world [29], and (2) student-centered teaching: The student becomes the protagonist of the teaching-learning process [7]. In this way, hybridization appears as an innovative trend capable of increasing the benefits and possibilities of individual pedagogical models [28].

Due to the situated learning context promoted by pedagogical models, such as Sports Education (SE), Cooperative Learning (CL), and TGfU, these facilitate their hybridization with other models [30], such as TPSR, due to the adaptability and interdisciplinarity that characterizes this model [21]. Empirical studies on hybrid longitudinal programs of the SE and TGfU models, developing different contents, such as soccer, tennis, badminton, softball, and volleyball, have shown significant improvements in the intention to be physically active, creating great sports adherence to improve healthy habits in the future [31] improving the affective, cognitive, and physical domains of the students [32]. This kind of hybridization could be useful to help teachers access a multi-model approach in their classes [33] that adapts to current educational frameworks [34].

Along these lines, there are other investigations that address the hybridization of the TPSR with the SE model, in didactic units from 16 to 26 sessions on content, such as Kickboxing and Xball [35,36], showing improvements in the values of responsibility, reduction of violence, improvement of competition and relationships between participants [37]. The TPSR has also been combined with other models, such as CL, where multiplier effects are expected with hybridization [38], and with active methodologies, such as Gamification (GF), observing its effectiveness to reduce demotivation and increase the self-determination index [39]. In short, there seem to be greater advantages when pedagogical models are applied in a hybridized way compared to when they are implemented independently, since the combination promotes obtaining better results in different domains, avoiding the limitation presented by individual models [28].

But it is necessary to consider that using new models can give new difficulties to the teachers, such as the time load required before, during, and after the intervention [40] or the difficulties encountered by the teacher when developing the methodology and adapting it to the educational curriculum [41]. Therefore, teachers rarely manage to implement the model in depth or over a long period of time [42]. If we focus on programs based on the hybridization of pedagogical models [28], Casey and MacPhail [33] state that teachers who used a multi-model approach were limited by their capacity to reconceptualize teaching with new changes. On the other hand, in a study based on SE and TGfU, the teachers in training found handicaps in its application, due to their limited experience and knowledge in this combination of pedagogical perspectives [43]. In fact, Araujo et al. [44] highlight the importance of education and training for those professionals who apply this methodology.

Despite the benefits of teaching based on model hybridization and the ease shown by TGfU and TPSR to be combined [21,28,30], no intervention studies have yet been found in which both pedagogical models have been hybridized [28]. In this sense, the combination of the TPSR with the TGfU is presented as a new pedagogical trend that aims to achieve a multilateral and comprehensive development of students through the improvement of different domains, such as motor, psychological, cognitive, affective, social, emotional, educational, and behavioral [20,25].

In addition, given the effect of both models to promote self-determined motivation in students [16,21], the present study follows the theory of self-determination (SDT) as a frame of reference [45]. This macro-theory establishes that the motivations follow a continuum of self-determination that goes from a state of demotivation to reaching intrinsic motivation through other motivational regulations, such as extrinsic sources of motivation [45]. According to Ryan and Deci [46], autonomous motivation represents the highest level of self-determination. Autonomous motivation is defined as engaging in activities for the interest and satisfaction derived from the activity itself, without any external contingency (e.g., rewards, praise). This kind of motivation is represented by intrinsic motivation (people perform the activity because they find it attractive and fun), and identified (people identify with the value of the activity and have a high degree of willingness to act) and integrated (people find the activity congruent with other values and interests in their life) regulations. On its part, controlled motivation is represented by external regulation (people act to obtain rewards or avoid externally imposed punishments), and introjected regulation (people act to obtain internal rewards, such as feeling good in case of success or to avoid anxiety and guilt in case of failure). The absence of motivation is represented by demotivation (people feel incompetent and uninterested). Additionally, within the self-determination theory, the micro-theory of basic psychological needs establishes the existence of three basic psychological needs (autonomy, competence, and relationship with others) that determine the state of health and psychological well-being of an individual [47]. In physical-sport activities, when people interact with their environment need to feel competent (feeling of mastery of the task), autonomous (feeling of being the initiator of one’s own actions), and related to others (feeling respected by others and desire to feel connected with them) [48]. Social environments that support autonomy (such as through the TPSR) provide the students the ability to improve their academic performance, be more creative and better adjusted, be more engaged in school, and feel less stress [49]. Autonomy support constitutes the key element to satisfy basic psychological needs [50], and thus, achieve higher levels of self-determination and intrinsic motivation [51].

Within the SDT framework, we find Vallerand’s hierarchical model of intrinsic and extrinsic motivation [52], which establishes that social factors, such as responsibility [53,54,55], can be considered as a trigger for the satisfaction of basic psychological needs (autonomy, competence, and relatedness) that in turn promote a more self-determined motivation in students [56], and generate positive consequences at the cognitive, affective, and behavioral levels [24], related to well-being. In this way, the hybridization of TPSR and TGfU is intended to foster student responsibility and promote a more self-determined motivation achieving greater health-related benefits.

Based on this theoretical foundation, the main purpose of this study was to know the effects of applying hybridization between the TPSR and TGfU model in first and second year Secondary Education students on the improvement of health measured through the intention to be physically active and the psychological variables of the students. Parallel to applying the hybridized program, the second purpose was to know the teachers’ perception about the implementation of the hybridized program, its advantages, difficulties, and proposals for the future. It is hypothesized that the TPSR and TGfU hybrid pedagogical model will improve the health habits and the psychological variables of the students. The teachers’ perception of this program will be positive, although some aspects will have to be modified to improve the model and make its implementation easier.

## 2. Materials and Methods

### 2.1. Design

It is a quasi-experimental pre–post study [57] where a mixed method research approach (Mixed Methods) [58,59] was applied with an embedded design of quantitative predominance [60]. To analyze the variables of this study, questionnaires were applied to the students before and after (pre- and posttest) the completion of the educational program. At the same time, part of the teachers’ intervention was videotaped and analyzed through observational analysis, assessing their implementation of the hybridized model and complementing it with the interpretation of the teachers’ perception at the end of the intervention through semi-structured interviews.

### 2.2. Participants

Participants belong to two public Secondary Education centers located in two towns in the Region of Murcia, with a middle socioeconomic level and similar in both centers.

The teachers who participated in the study were four career teachers with professional experience in the field of teaching, aged between 22 and 46 years, two of them were part of the experimental group (EG) and another two from the control group (CG). The EG teachers were subjected to a period of continuous training (specified below) to carry out the implementation of the educational program.

The sample of students was selected for accessibility and convenience, and it originally consisted of 139 students between the two educational centers corresponding to first and second year of Secondary Education. The exclusion criteria that were established were the following: (a) That the student had completed the test that was carried out before and after the intervention, (b) that he completed all the scales presented in the booklet that contained the questionnaires, (c) to fill in the questionnaire correctly according to the explanations that were made during the process. After applying these criteria and homogenizing the sample, atypical cases were detected using the Mahalanobis distance. The final sample consisted of a total of 99 students (51 girls and 48 boys) aged between 12 and 14 years, the mean age being 12.63 (SD = 0.72). Of these subjects, 55 belonged to the CG and 44 to the EG.

The age and sex distribution were similar in the control and experimental groups. Regarding the teachers involved, none had previous experience in applying an educational program based on pedagogical models in an ordinary classroom. However, prior and continuous training was carried out for the teachers who intervened in the EG, by monitoring teaching behaviors through video recordings and subsequent analysis by qualified specialists in the field of observational methodology; these aspects are specified in the following sections of the method. 

### 2.3. Mesures and Instruments

Given the nature of the research, based on Mixed Methods, several instruments of a quantitative and qualitative nature were used to verify the behavioral changes of the student and teacher participants involved.

#### 2.3.1. Questionnaires for Students

In the present study, several questionnaires previously validated and adjusted for Secondary Education students were used. The questionnaire sheet had a total of 59 items with a completion time of 25 min, previously tested by the researchers involved in the study.

(1) Personal and Social Responsibility Questionnaire (PSRQ). The validated Spanish version [61] of the Personal and Social Responsibility Questionnaire [62] was used. The questionnaire includes 14 items grouped into two factors: Personal responsibility (for example—I make an effort) and social responsibility (for example—I respect others). The participants answered the test with a Likert-type scale between grades 1—totally disagree and 6—totally agree. The test instructions were presented at the beginning of the test accompanied by the following sentence: “We are interested in how you normally behave during physical education class. There are no right or wrong answers. Please answer the following questions honestly and circle the number that best represents your behavior”. The Cronbach’s alpha obtained in the pretest and posttest in social responsibility was 0.72 and 0.71, respectively, and 0.78 and 0.75 in personal responsibility.

(2) PLOC Motivation Questionnaire. The test “The Physical Education Motivation Questionnaire” [63] was used. This questionnaire includes 20 items grouped into five factors: Intrinsic motivation (for example—because physical education is fun), identified regulation (for example—because I want to learn sports skills), introjected regulation (for example—because I want the teacher to think that I’m a good student), external regulation (for example—because I’ll have problems if I don’t), demotivation (for example—but I don’t really know why). The participants answered the questions on a Likert-type scale between grades 1—totally disagree and 7—totally agree. The questionnaire began with the following sentence: “I participate in physical education classes…”. Regarding the reliability of Cronbach’s alpha, values above 0.70 were obtained in all dimensions. In addition to the factors that have been discussed, two new study variables were also included: Autonomous motivation with an internal consistency of 0.80 in the pretest and 0.82 in the posttest, which was calculated with the formula: (intrinsic motivation + motivation identified)/2 and controlling motivation with 0.82 in the pretest and 0.84 in the posttest, which is found with the formula (introjected regulation + external regulation)/2 [64]. On the other hand, the self-determination index (SDI) was also found [52].

(3) PNSE Basic Psychological Needs Questionnaire. A version of the Spanish test validated for the educational context [65] was used, based on the Basic Psychological Needs in Exercise Scale [66]. It includes 12 items that are grouped into three subscales: Autonomy (for example—we do things that are of interest to me), competence (for example—I think I improve even in tasks that my colleagues consider difficult), and relationship (for example—relations with my classmates are very friendly). The participants answered the test questions in a range from 1—totally disagree to 7—totally agree. The questionnaire began with the following sentence: “In physical education classes…”. Regarding the validation of the instrument, the subscales obtained the following internal consistencies (Cronbach’s alpha) for autonomy, competence, and relationship in the pretest were, respectively: 0.69, 0.78, and 0.56 in the posttest: 0.73, 0.92, and 0.77. In addition, the Psychological Mediators Index (PMI) was used as a variable that obtained an internal consistency of 0.74 in the pretest and 0.84 in the posttest.

(4) Sport Satisfaction Instrument SSI Questionnaire. The validated Spanish version [67], based on the Sport Satisfaction Instrument (SSI) [68], was used. This instrument is composed of eight items grouped into two factors: satisfaction/enjoyment (five items), (for example—I usually have fun in physical education classes) and boredom (three items), (for example—in the physical education class, I hope it ends quickly). The participants answered the test questions on a Likert scale ranging from 1 (totally disagree) to 5 (totally agree). The heading of this test began with the sentence: “Indicate your degree of disagreement or agreement with the following statements, referring to your physical education classes.” Regarding the internal consistency of Cronbach’s alpha for the dimension referring to satisfaction or enjoyment was 0.76 (post = 0.78), and Cronbach’s alpha for the boredom dimension was 0.83 (post = 0.61). This last value is lower than 0.70, which is the cutoff criteria to be considered acceptable. However, some authors [69,70], consider values above 0.50 as acceptable in scales with fewer items.

(5) Questionnaire of intention to be physically active. The validated Spanish version [71], based on the Intention to be Physically Active Scale [72], was used. It includes five items grouped around a single factor (for example—I am interested in the development of my physical form). The participants answered following a Likert scale between grades 1 (Totally disagree) and 5 (Totally agree). The internal consistency of Cronbach’s Alpha for the pre and post was 0.70 and 0.75, respectively. 

In summary, Cronbach’s Alpha was analyzed, achieving good values for all scales, positioned above 0.70, and only one dimension, Sport Satisfaction Instrument, was lower (0.61), being above 0.50, which is the minimum for those scales with fewer items [69,70].

#### 2.3.2. Instruments for Teachers

(1) Tool for Assessing Responsibility-Based Education (TARE). The validated Spanish version [73] of the original scale was developed by Wright and Craig [74]. Of the totality of the tool, only the first section was used that refers to the registry of strategies used by the teacher to promote responsibility, equipped with a system of nine categories: (1) Model of respect (M); (2) set expectations (E); (3) gives opportunities for success (S); (4) promotes social interaction (SI); (5) assign tasks (T); (6) leadership (L); (7) granting of choice and voice (V); (8) role in evaluation (A); (9) transfer (Tr). Because the hybridized methodology contemplated both the TGfU model and TPSR, three new observation categories were included in the TARE instrument related to the TGfU model, to guarantee the behavioral fidelity of this intervention: (10) Modified sports game (J); (11) tactical awareness; (12) execution of the skill (SE). These categories were constructed based on the specific strategies of the model and session structure carried out in TGfU [75]. For the adaptation of this instrument, the procedure of Gil-Arias et al. [76] where the main researcher and a teacher with experience in pedagogical models observed more than 12.5% of the sessions [77], reaching a 100% interobserver agreement in those categories that had been included (10, 11 and 12). In this way, a new observational instrument called TARE + TGfU was obtained.

(2) Semi-structured individual interview. The instrument has 14 questions adapted from the study carried out by Manzano-Sánchez et al. [78]. The EG teachers underwent these interviews individually at the end of the intervention. The questions referred to their perception of the methodology itself (differences in student behavior, content/students more likely to receive the methodology), the usefulness of the training carried out, perception of hybridization, and a series of questions about the possible advantages/disadvantages and if they would apply some of the methodologies used in the future.

### 2.4. Procedure

First, the scientific literature was reviewed to assess those instruments that best suited the type of study and intervention to be carried out, deciding the study variables.

Subsequently, the research design and the selected instruments were submitted to the criteria of the Ethics Committee of the University of Murcia (2680/2019) to ensure that they complied with the Helsinki Declaration guidelines on research ethics.

Once the approval of this commission was obtained, a report was written to the management team and teachers of the centers involved in the study, explaining the details of the investigation. Subsequently, authorization was requested from the educational centers and the parents/guardians of the participating students.

#### 2.4.1. Hybrid Intervention Program TPSR + TGfU

The TPSR hybridized with the TGfU was implemented for 11 weeks (two sessions of 50 min per week) in both educational centers where the main teaching strategies were applied. The contents of the educational programs of the center, which are governed by current Spanish educational law, were followed [79]. The contents that were developed in this study were related to multisport: Initiation to basketball, futsal, and volleyball. Objectives and contents for every week (both groups), principles and tactical problems, skill-execution task examples for TGfU, as well as levels, strategies, and task responsibilities for TPSR, have been detailed in Table 1.

During the intervention program, a schedule was established where the sessions that were to be recorded were established to provide feedback to the teachers in charge of developing the intervention program and the sessions that were to be recorded to obtain data for the present study. The analyzed sessions were a total of 13, of which 6 of them belonged to the experimental group (EG), 3 of each of the teachers, and 7 analyzed sessions of the control group (CG).

Considering how the sessions were carried out with this hybridization and based on the TPSR proposal [18], the sessions adopted the following structure: (1) Awareness, the teacher establishes the level of responsibility in which his students are situated and reflect on everyday examples involving the achievement of this level; (2) responsibility in action, where the main part of the session takes place, developing the strategies that contribute to promoting levels of responsibility; (3) group meeting, a group meeting is held to assess the session and reflect on the involvement and participation of the students and the progression of the group in the level of responsibility that is being worked on and (4–5) self-evaluation and peer evaluation, where students perform an evaluation of their own performance, that of the class in general and that of their teacher. Within the structure of the TPSR, the hybridization with the TGfU model was carried out, which took place in part (2) of the TPSR, responsibility in action. In it, an adaptation of the work of Kirk and MacPhail [80] is applied, (1) game forms, it is a modified global game that aims to have a first contact with tactical elements typical of sport (2) tactical awareness, once the students detect problems to achieve the objective in the game, the teacher brings the group together and through guided questions, tries to orientate their students so that they offer a solution to the problem posed (3) execution of the skill, a modified game by exaggeration or simplification is proposed aimed at improving the technical-tactical problem that the students themselves have detected through awareness and (4) repetition of the game forms or evolution of the same, the same situation is proposed or an evolution of it, so that students can put into practice what they have learned through reflection and practice of modified games [14]. During the group meeting (third part of the session structure of the TPSR proposal), reflective aspects related to the TGfU were carried out.

Regarding the strategies implemented by the teachers, which had to be carried out at least once in each session, we found the following actions contemplated in the TARE + TGfU instrument. Specifically, those related to TARE were those that we detail below; be an example of respect, remembering the importance of punctuality; respect everyone’s turn of participation without interfering; set expectations at the beginning of the session, indicating the objectives of the session; work on the level of responsibility which is expected of them; give opportunities for success, putting different levels of difficulty in activities or helping those who have more difficulties to achieve their success; promote social interaction, with cooperative activities; promote teamwork or problem solving; assigning tasks by distributing responsibilities, such as material manager or warm-up director; attribute leadership by assigning group leadership roles, such as coaches or captains; give opportunity for choice and voice, listening and taking into account the opinions of the students throughout the session and especially in the final debate and self-evaluation; transfer, especially at the end of the session, indicating what the worked value is for, which may have repercussions outside the classroom; respect the partner’s turn, to listen to their opinions. All these strategies are primarily intended to increase the levels of personal and social responsibility, although they also contribute to support student autonomy.

In relation to the TGfU, the strategies implemented by the teachers in the PE class were, according to the game forms, posing situations that modify the real game by the following actions; exaggeration, changing the width or length of the field; simplification, reducing the standards to be met to achieve the objective; tactical awareness, proposing to the students that they reflect and decide on the solution to the problem; representation of a specific game situation that has been graphed; execution of the skill, by solving technical-tactical situations of opposition-collaboration (1 × 1, 2 × 3 or 3 × 2). The latter could be carried out with specific instructions by the teacher, for example, making a shot to or from a certain area to send the ball to a place far from the opponent’s reach, etc. All these strategies are intended to improve the social environment for autonomy support, competence, and relatedness perceptions.

#### 2.4.2. Continuous Training of EG Teachers

To implement any type of educational program, specific professional development of teachers is necessary [25]. For this reason, the two EG teachers who applied the hybridized methodology were trained using a supportive two-phase approach [81]:

(1) Previous or initial training: Two instructional courses were held (5 h in TPSR and 5 h in TGfU) where the global and specific strategies of each teaching methodology and how to approach hybridization were explained. In addition, they were provided with a “model guide” to review the strategies. Previously to the implementation, teachers had not used any of these two methodologies in their classes, and they did not know the aspects of TPSR or TGfU. This training taught the teachers to incorporate this hybridization in their classes. However, because they did not have experience, continuous training was necessary to ensure that the implementation was adequate. 

More specifically, TPSR training included how to integrate the session structure in their classes (awareness, activity plan, group meeting, and self-assessment), the five levels of responsibility, some specific strategies to be developed in the classroom (e.g., modifying the rules, redefining success, personal work plan, group objectives, and community services), and conflict resolution with strategies like five clean days (for individual conflicts) or the dialogue bench (for group conflicts). TGfU training consisted of knowing the session structure (game form, tactical awareness, and skill execution), learning games and game forms to develop the tactical awareness, tasks, and predesigned lessons for practice, analysis, and discussion to be able to design the final version of the learning unit to be implemented. Finally, teachers learned to gather both methodologies in the same lesson following this sequence: Awareness, game form, tactical awareness, skill execution, game form, group meeting, and self-assessment.

(2) Continuous training: In a sequential and progressive way, teachers received feedback, support, and guidance in applying the TPSR and TGfU models. Specifically, an expert in pedagogical models oversaw regularly attending the preparatory sessions of these two teachers of the experimental group to monitor and advise, recording their performance; later, in a follow-up meeting, feedback was given to these teachers in charge of implementing the program. The expert was a university teacher that had used these two methodologies in the last year with experience of more than 10 years in this field and who was advised by another three experts in active methodologies.

Subsequently, the principal investigator intervened in the preparation of the first sessions of the program to ensure that the main characteristics of the hybrid pedagogical models were developed. Content timing, examples of activities and progressions from the hybridization approach, examples of action situations, guidelines to be followed in each session progressively, and clarification of doubts were proposed.

During the intervention, this continuous training was followed with its follow-up: On each second week of recording, a session of each teacher was chosen, and the main researcher analyzed it with the TARE + TGfU instrument, recording the strategies that were approached during the intervention in intervals of 5 min [82]. He used the TARE + TGfU instrument (explained in Section 2.3.2) to give feedback to the teacher. Seven days later, the main researcher himself carried out a second exhaustive analysis of the recording, from there a document was prepared with those aspects that were being developed correctly to encourage the teacher to continue carrying them out and proposals for improvement, tips, or next objectives for the teacher to take into account in the next sessions were also offered. In addition, a meeting was held with each of the teachers in the training of the EG group to comment on the aspects that appeared in the drafted document. This same procedure was repeated throughout the intervention, once feedback was given to the teacher, a week was left for the teacher to apply the new guidelines, and in the second week, the same process was repeated to know its evolution. In addition, at the end of each session, teachers were encouraged to assess their performance, and the session was held with the students to encourage their own reflection on the implementation of the model.

The students were progressing in the different levels of responsibility throughout the 11 weeks. However, level 5, which dealt with the transfer outside the educational setting, was present as a goal to be achieved throughout the intervention.

#### 2.4.3. Loyalty of the Hybrid Program Registry

Before carrying out the intervention in this study, the principal investigator was instructed to make an adequate record of the teacher’s performance in relation to the dimensions described previously [83]. Finally, interobserver reliability of more than 80% was achieved, which allowed the analysis of the sessions to begin. It was calculated with the following formula: AT = TA/A + D (total agreement = AT; total agreements = TA; agreement = A; disagreement = D) [84].

The video observation of the EG and CG teachers was used to collect information through the TARE + TGfU instrument. A total of 13 sessions were recorded, the average duration of the sessions was approximately 50 min. In addition, each session that was recorded had the participation of an expert in the methodology who oversaw carrying out the first analysis with this tool, viewing in situ the session. Later, after a week, that same expert carried out another analysis of the same session to see if it exceeded 80% reliability, thus achieving intra-observer reliability.

### 2.5. Data Analysis

For data analysis, version 24 of the SPSS statistical analysis program was used. As for the qualitative results, they were analyzed using the ATLAS ti program. V. 7.1.3 obtaining a family of codes of positive and negative perceptions and a count of those codes that referred to the same dimension. 

First, the reliability of the different variables measured in the different tests was calculated, the dimensions of variables were created to be able to work with them later during the data analysis. Descriptive statistics were obtained for all the dimensions that were the object of study, correlation analysis, and internal consistency with Cronbach’s Alpha coefficient.

The vast majority of the coefficients exceeded the 0.70 reliability values that are considered acceptable for psychological scales [85]. The only dimensions that were below these values were the pretest of autonomy with 0.69 and the posttest dimension of boredom with a 0.61 value, both are defined as acceptable according to Sturmey et al. [86]. Finally, the pretest dimension of psychological needs referring to the relationship had 0.56 of internal consistency, which is defined as valid if it is a scale of few items [69].

Subsequently, to achieve a more complex analysis of the intervention, and thus, know the effect of the implementation, homogeneity tests were performed using a multivariate analysis of repeated measures (MANOVA) of the different variables according to time and group. To know the results of this hybridized program, the pre and posttest results of both the control group and the experimental group were considered, observing the significant differences in time between both groups. 

## 3. Results

### 3.1. Results of the Strategies Used

In order to be able to know the validity of the implementation of the educational program and to verify if the teachers of the EG or the CG were applying in an ideal way the methodology of this hybridization of models, the strategies used with the TARE + TGfU were evaluated (Table 2).

The descriptive analysis reflected that the EG had higher values in the U of Mann Whitney in all the strategies of the models and reflected significant differences in favor of the EG in: “providing opportunities for success”, “fostering social interaction”, “granting of choice and voice”, “role in evaluation”, “transfer”, “tactical awareness” and “skill execution”. For this reason, the EG teachers were able to improve the teaching strategies for the most part based on a constructivist perspective.

In addition, it is important to highlight how the CG reached 0.00 mean values in some variables, such as “role in evaluation”, “transfer”, “tactical awareness”, or “skill execution”, due to using more analytical strategies focused on technical learning, which deprived students of tactical situations of meaningful sports learning. 

### 3.2. Results of the Interviews

The analysis of the interviews revealed a total of 15 codes that were ordered according to the number of extracts that were selected within each of the codes. In turn, the codes of both teachers were included in the so-called Code families, which turned out to be a total of three families (Figure 1).

For the development of this section, it was started from each of the families: “Model of Personal and Social Responsibility”, “Comprehensive Teaching” (TGfU), and “General concepts and suggestions” (Figure 1) to describe the results:

Teaching personal and social responsibility: The five codes of this family were considered positive, highlighting the positive assessment with 15 extracts indicating aspects as indicated by teacher B “Following the answers from before a bit, on the one hand, yes, the Responsibility model was sufficiently adjusted to the needs of the students”. In the same way, other codes were also valued, such as the transfer of values or the promotion of values, as indicated by teacher A “as students who are new to the institute, it was good for them to reinforce the values with the different levels of responsibility, especially because of their preadolescent age, it can help them a lot to deal with a social context, whether with family, friends or teachers”. In addition, the codes of “comfort applying the model” or the future application of the same were formed as teacher A says, “I would still use the Responsibility model in the future”, speaking exclusively of the TPSR.

Teaching games for understanding: This family was made up of three codes, considering the one with the most extracts being the negative response from the students, since this code was only referred to by both teachers when they spoke of this methodology with 12 extracts, thus, Teacher A said that the first weeks were the most difficult “especially the first weeks were the most insecure because I saw that it did not work very well with the students because it was a new model for them and for me”. On the other hand, it had a neutral code called “influence of the content when applying the model”, where some contents were more optimal than others, and a positive code “positive assessment” (comprehensive teaching) with six extracts as indicated by teacher A “As for the comprehensive model, since they were clear about the structure of the session as the sessions progressed, it helped the students to have a better attitude and behavior”.

General concepts and suggestions: Including aspects of teacher training and those that made reference to both models. The one about “problems applying models” stands out, with 16 extracts, especially indicating as teacher A says that “you have to invest a lot of time in the explanation and the time of physical activity is quite reduced”. Although this was the only code that was negative, since the rest were considered neutral (suggestions for improvement and importance of the experience for practice). On the other hand, the positive aspects involved codes, such as “Improvement compared to traditional teaching”, which is very important for using models, as stated by teacher B “With another less structured and more traditional model one can achieve great motor commitment in some occasions, but in others, there may be greater lack of control”. Receiving adequate training during the course and the importance of continuous training “This feedback helped me to redirect a bit my idea of the models and the approach that I was giving to it. I noticed that it was useful and that it helped me to improve as a teacher within that model and to be able to teach it with guarantees” (teacher B), in addition to the positive response from the students, “I could observe that many girls who tend to be less participatory became more involved” (teacher B), thus improving the sports adherence of the students who practiced less physical activity and directly improving their healthy habits.

### 3.3. Results of the Inferential Analysis

To assess the effects of the intervention program in each group, a repeated measures MANOVA was carried out. The results show that there are significant differences in the intrasubject factor Time (Wilks ‘Alpha = 0.688, F (3.255) = 12, *p* = 0.001) and they remain very close at the level of the Group factor (Wilks’ Alpha = 0.796, F (1.842) = 12, *p* = 0.054). However, there are no significant differences in the Time × Group interaction (Wilks’ Alpha = 0.844, F (1.326) = 12, *p* = 0.219).

These results were then analyzed at the univariate level to observe those variables that presented significant differences. For the intergroup differences, significant differences were observed in the EG group for the time factor in the variables of autonomous motivation, their motivation with the school and their active intention to practice sport or physical activity (F = 10.943, *p* = 0.001), demotivation (F = 6.667, *p* = 0.011), SDI (F = 6.509, *p* = 0.012), relationship (F = 12.958, *p* = 0.001), autonomy (F = 5.488, *p* = 0.021), IMP (F = 7.026, *p* = 0.009), enjoyment (F = 12.913, *p* = 0.001), personal responsibility (F = 4.341, *p* = 0.040), social responsibility (F = 7.963, *p* = 0.006), and intention to be physically active (F = 9.037, *p* = 0.003). For this we compared the change between pre- and posttest in both groups, the intervention got an improvement in most aspects of the study, increasing all cited variables except demotivation, which decreased.

Table 3 shows the mean and standard deviations of the different variables in the pretest and in the posttest, differentiating between groups. The p-values obtained when comparing these estimated means (using the Bonferroni correction) are also included. No differences were found in the pretest, but in the posttest some variables increased in the EG group, more specifically, autonomous motivation (*p* = 0.000), SDI (*p* = 0.011), and intention to be physically active (*p* = 0.029). In this sense, using this methodology allowed the EG to improve.

On the other hand, when comparing the variables between the pretest and the posttest for each group, it can be seen that there are only significant differences for the relationship variable (*p* = 0.035) in the control group. However, in the experimental group, the scores of the variables autonomous motivation (*p* = 0.000), SDI (*p* = 0.005), relationship (*p* = 0.004), autonomy (*p* = 0.017), PMI (*p* = 0.004), enjoyment (*p* = 0.001), personal responsibility (*p* = 0.015), social responsibility (*p* = 0.001) and IPA (*p* = 0.000) increased, while demotivation has decreased (*p* = 0.017). For this reason, CG did not show any differences with other variables except relationships. On the other hand, we checked that EG improved most of these variables again.

## 4. Discussion

The objectives of this study were, on the one hand, to know the effects of applying hybridization between the TPSR and TGfU model in secondary school students for the improvement of health through the intention to be physically active, as well as responsibility, motivation, the satisfaction of basic psychological needs and satisfaction in PE. Second, to know the teachers’ perception about the implementation of the hybridized program, advantages, difficulties, and proposals for the future.

Regarding the first of the objectives, the results reflect statistically significant differences in the more self-determined motivation and the intention to be physically active among the participating groups, in favor of the experimental one. These results are in line with those found by Gil-Arias et al. [31], that after applying a program based on the hybridization of the Sports Education (SE) model and the TGfU for 16 sessions, an increase in motivation and intention to be physically active was observed in students. Following Merino Barrero et al. [24], the improvements that appear in these variables leave evidence of the contribution of this educational program to the health of adolescents, since responsibility, basic psychological needs, and self-determined motivation predict the intention to be physically active and healthy lifestyles. In other studies where TPSR is applied, it is observed that there is improvement in health education. In this sense, Melero-Cañas et al. [87], show the effectiveness of the hybridization of pedagogical models, in this case, TPSR and GF, to promote a more active lifestyle in adolescents. 

The results also show statistically significant differences in the experimental group over time in the variables of autonomous motivation, IAD, IMP (especially due to an improvement in autonomy and social relationships), enjoyment, personal responsibility, social responsibility, and intention to be physically active, while demotivation was significantly reduced. Different empirical studies [21,39,88,89], which applied the TPSR independently, in different areas and educational stages, show the applicability and effectiveness of this pedagogical model to increase motivation, basic psychological needs, personal and social responsibility, the school climate, and the intention to be physically active in the students.

Taking SDT as a reference and in line with Vallerand’s hierarchical model, it can be observed that applying hybrid methodologies that manipulate social factors, such as responsibility, are suitable for producing an improvement in basic psychological needs and promoting a more self-determined motivation [19], generating positive consequences at an affective and emotional level, such as enjoyment and the intention to be physically active, in line with the studies by Manzano-Sánchez et al. [54], Menéndez-Santurio and Fernández-Río [90], and Merino-Barrero et al. [24].

In relation to the TGfU, there are practically no studies that have analyzed the effects of this pedagogical model on psychosocial variables, although there are studies that investigated its effect on the intention to be physically active [31], and on levels of physical activity, such as Wang and Wang [17], where they found that students who received an education under the TGfU achieved higher levels of physical activity than students who received conventional education. In conclusion, its implementation enhances the promotion of intense physical activities and allows achieving the recommended time for moderate-vigorous physical activity in PE classes.

On the other hand, the results of the TARE + TGfU instrument show significant differences in favor of the teachers of the experimental group in those items that are part of the strategies of hybrid pedagogical models, such as the granting of voice and vote, the role of evaluation, transfer, tactical awareness, or skill execution. These results follow the line of different studies [21,82] that found significant differences between teachers who applied directive teaching and those who implemented pedagogical models independently or hybridized [32,35,43].

Considering the second of the objectives of the present study, it is observed that the teachers perceived hybridization as an alternative capable of producing improvements compared to a conventional methodology, thanks to the session structure offered by both models independently [78,91], and that could be enhanced with its combination. In this sense, the teachers also highlighted the positive response of the students to the methodological approach offered in each session, results like those found by other authors who hybridized the CL with the TGfU [92] or the SE with the TPSR [35,36].

Other statements made by teachers are related to the positive assessment of both models they have independently. On the one hand, they highlight the ability of the TPSR to promote values and transfer them to different contexts of daily life, showing its comfort when implementing it and specifying a possible future application, aspects that are collected in the interviews carried out in Manzano-Sánchez et al. [19] and Manzano-Sánchez and Valero-Valenzuela [88] studies. On the other hand, they indicate that the TGfU favors the performance and behavior of the students when the intervention progresses, thanks to the establishment of a permanent routine during the sessions. Empirical studies that applied the TGfU [16,93,94] found progress of the students towards more positive attitudes, achieving greater participation, enjoyment, and motivation in the proposed tasks, which lead to greater physical, emotional, and psychological well-being.

However, there are also manifestations that refer to perceived difficulties when applying the hybridized methodology, above all, due to the time dedicated to the explanation at the beginning of the intervention. The teachers who participated in the study by Gutiérrez et al. [91], based on the hybridization of SE and TGfU, indicate this same problem. For this reason, the teaching experience when implementing pedagogical models is a factor that plays a fundamental role [82], especially when it comes to hybridization [28,43]. According to the teachers, the difficulties found in the present study are counteracted with detailed initial training and continuous training that allows them to receive feedback capable of offering a solution to the problems that appear in practice.

Thus, teacher training is also considered an essential component in using pedagogical models [25,28]. Finally, the teachers also state that the content influences when applying the models, although this is one of the limitations that is intended to be solved with the hybridization of pedagogical models, enhancing the benefits that each one of them can provide [95], and covering its implementation to a greater number of contents in PE [28]. 

Scientific evidence shows the ability of TGfU to improve motor, cognitive and affective learning, as well as that of TPSR, to produce positive results in different aspects, such as social, psychological, emotional, educational, and behavioral [28], benefits that remain when it hybridizes to SE [37,96]. However, these positive effects do not occur when the variables have a physical, motor, or cognitive component [97], being able to alleviate with the combination of the TPSR with other pedagogical models, such as the SE [36] or the GF [98], whose combinations have yielded positive results in motor domains, possibly due to the influence of SE on students’ perception of competence, and autonomy, and due to the influence of GF on student participation and involvement. However, despite the influence of TGfU on levels of physical activity [17], motor skills and technical-tactical elements [25,28], and the influence of TPSR on psychosocial aspects, there is still no scientific evidence of hybridization of both pedagogical models, this article being the first experience of their combination. Therefore, a new research trend could be constituted in the field of PE, with the sufficient capacity to produce improvements and positive consequences in the health of young people at a physical, social, and psychological level.

The main limitations of this study are the short intervention time in which the educational program was implemented, since teachers new to the methodology expressed the little adaptation period they had to be able to apply it. On the other hand, there were only two teacher participants in the EG, and the whole sample was composed of 99 students. The type of sampling has been intentional for accessibility. This is not a representative sample in which all individuals were equally likely to have been selected. Future studies that address this issue should be carried out using a sample with greater methodological validity, such as random, strata, and/or cluster samples. Instruments used to measure these variables are based on questionnaires; therefore, they collect opinions, but not real behaviors or values. One of them is about their intention to be physically active, and it is very likely that the levels are lower than they state because of social desirability. In addition, it would be interesting to control the confounding factors to know the previous methodological approach that EG teachers used before the intervention, to know which were the strategies that they acquired thanks to the training received and which they had previously. It would also be interesting to include other instruments that allow us to know if there is a significant improvement in the levels of physical activity achieved by students after having applied to this educational program. This study should be taken as exploratory research considering the responsibility a trigger of the social aspects, because of that, future studies should be developed to check these preliminary results. Finally, we intend to address future studies by hybridizing these two methodological approaches with new curricular content to obtain more evidence about this hybridization of models in the school context.

## 5. Conclusions

In conclusion, considering the theoretical background of the self-determination theory, the situated learning, and the student-centered teaching in the PE context, the hybridization of the TPSR with the TGfU is presented as a methodological alternative capable of helping to improve certain limitations arising from applying a pedagogical model independently, showing its effectiveness in increasing self-determined motivation, basic psychological needs, enjoyment, personal responsibility, social responsibility, and the intention to be physically active. In this way, the combination of both pedagogical models can contribute to students having a more active lifestyle and acquiring healthier life habits.

In addition, teachers perceive this innovative proposal positively, highlighting its nature to organize and structure the session, an aspect that generates greater comfort when applying the methodology. This event allows teachers to consider implementing hybridization in the future, also thanks to the positive response of the students. However, its application requires initial and continuous training by experts to reduce the possible difficulties that may appear during practice, teaching experience being a determining factor.

## Figures and Tables

**Figure 1 children-08-00573-f001:**
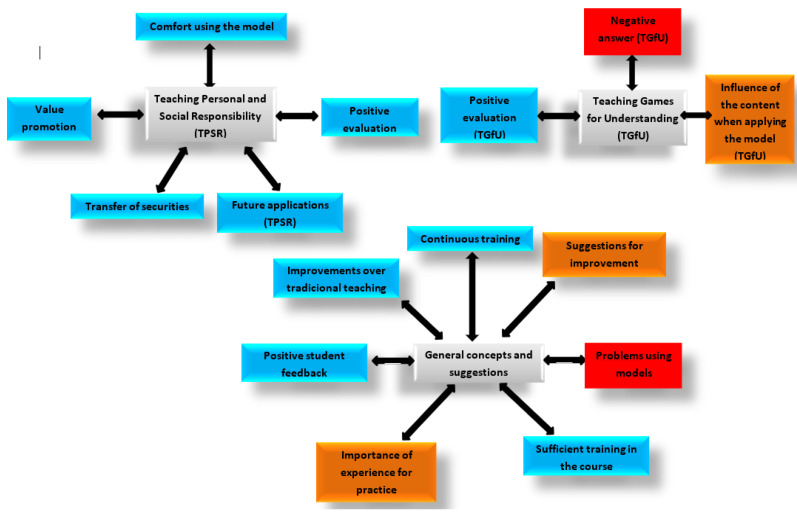
Teacher evaluations on the implementation of pedagogical models in physical education.

**Table 1 children-08-00573-t001:** Weeks, objectives, contents, principle, tactical problems, and skill-execution tasks, levels of responsibility, strategies, and examples of tasks during the intervention.

Intervention Week	Objective	Content (CG)	Content (EG)	TGfU (Principle and Tactical Problem)	TGfU. Skill-Execution Task Example	TPSR (Level of Responsibility: Strategies)	TPSR. Example of Responsibility Strategy
Week 1	Initiate students in futsal.	Futsal: Ball control, passes, and shots.	Futsal: Decision making with the possession of the ball.	Keep possession of the ball Who to pass?	In groups of six students, they form a circle (rondo), and one student stands in the middle trying to take the ball while making passes.	Level 1: Class norms were established in consensus with the students.	The students made a mural showing the rules of class behavior that they made themselves place on the wall of the pavilion.
Week 2	Understand the technical and tactical aspects of futsal.	Futsal: Shots on goal and opposition situations 1vs1.	Futsal: Decision making with the possession of the ball.	Advance with the mobile and invade the opponent’s field. Pass or advance?	Three students attack and two defend; the objective of the activity is to reach a goal line by passing the ball to the player who is free without resorting to dribbling.	Level 1: Teams were randomly configured to work on respect among students.	A discussion circle was held in the final part of the session where students had to adopt respectful attitudes, such as: Raising their hands before speaking or respecting their turn to speak.
Week 3	Know aspects of futsal defensive and offensive tactics.	Futsal: Defensive and offensive situations with numerical superiority and inferiority.	Futsal: Defensive tactical situations	Avoid marking. Where should I stand?	Two students defend and two attack; the students who defend the goal, must prevent the attackers from having a shot because there is no goalkeeper.	Level 2: Success was redefined in the different activities to encourage participation and effort	Activities were proposed where the result of the competition was not taken into account, but also the effort made by the group or the student, valued by the teacher and the rest of the classmates.
Week 4	Learn complex tactical aspects.	Futsal: Real game situations.	Futsal: Movements without the ball: lose the marks.	Attacking without the ball What is the best space to move to?	In groups of three, the students had to pass the ball to their teammates who came running to an area delimited with cones near the goal to get the point.	Level 2: We worked with an intensity scale, so that the students could measure their degree of effort from 0 to 10.	The students had a card during the activities they carried out this week to mark the perceived effort at the end of the activity from 1 to 10.
Week 5	Start the students in volleyball.	Volleyball: Serve, forearm pass, and setting.	Volleyball: Making a decision between the different types of hitting	Control the ball What attack helps us the most to place the ball on a partner?	In pairs, the students decide what type of attack to use to achieve the greatest number of hits without the ball falling to the ground.	Level 2: A modification of the tasks was carried out depending on the group that was working on them to promote participation.	The students with greater motor competence had to achieve a greater number of attacks at a greater distance, while for the rest of the groups, the objective was to overcome themselves during the activity.
Week 6	Know regulatory aspects of volleyball.	Volleyball: Collective games to develop technical-tactical aspects.	Volleyball: Regulation, rotations, and scoring.	Know the rules of the game What are the different field areas?	A reduced game situation was carried out (3 vs. 3) where a student assumed the role of referee to be able to explain the regulatory aspects of the game to his teammates.	Level 3: A distribution of tasks was carried out, the teacher explains the activity, but it must be the students who organize themselves independently to carry out the different roles and rotate. (coach, referee, player).	A game simulation was carried out, and the students themselves had to assume the roles of referee, player, coach, physical trainer, and scorer.
Week 7	Learn to work as a team in volleyball.	Volleyball: Reduced game situations and team competition.	Volleyball: Team communication, modified game situations.	Communicate action How should we communicate during the play?	Groups of four. One player serves from the other side, the receiving player shouts “mine”, and the placing player says the name of the player to whom the ball is going so that he is ready and passes it to the opposite field.	Level 3: Independent work of the students was promoted, establishing a series of game problems to which they had to solve and offer activities to be able to develop them.	The teacher played a game where the students who had brilliant ideas to provide solutions to the problem raised had a reward.
Week 8	Initiate students in basketball.	Basketball: Passes and shots.	Basketball: Decision-making between the different types of passes.	Keep possession of the ball. What type of pass to make?	In pairs, the students made passes between them, while moving laterally, choosing the type of pass depending on the distance that separates them.	Level 3: A personal work plan was provided, the students had a card that indicated the instructions for the task.	One student was in charge of reading the task to their group, and the rest were in charge of organizing the task independently, without resorting to the help of the teacher.
Week 9	Know defensive tactical aspects of basketball.	Basketball: Layup, three point shots, and blocks.	Basketball: Defensive tactics and markings.	Regain possession of the ball. How should I defend?	2 vs. 1, the objective of the activity is for the defending student to prevent the pass to the teammate who does not have the ball.	Level 3: During this week, the students continued to use the personal work plan strategy.	A student-teacher was chosen, who followed the instructions of the work plan, he was in charge of explaining the activities to the rest of the class.
Week 10	Selecting the shots to the basket.	Basketball: Specific roles, functions, and movements.	Basketball: Cooperation-opposition played situations	Achieve to shoot and score. From where can I be more effective?	A game is played to a 3 vs. 3 basket, the attacking team must shoot from defined areas so that the basket scores double.	Level 4: Students set some group goals and helped each other to achieve them	The students proposed additional objectives to each activity, such as: Pass the ball to all the classmates. In addition to the objective of the activity, the students helped each other to achieve this group objective.
Week 11	Solve situations of cooperation-opposition in basketball.	Basketball: Cooperation-opposition games and matches.	Basketball: Strategic principles of attack and defense, blockades.	Avoid marking. Where should I stand?	Basketball game 5 vs. 5 to a single basket, students who block an opponent’s shot achieve the same score as a basket.	Level 4: Reciprocal teaching was carried out, the students with the greatest experience in sport helped their classmates to improve in certain aspects.	Group captains were chosen to teach their teammates technical and tactical aspects of basketball during the activities, leaving a “time out”.

Note: CG = Control Group; EG = Experimental Group; Level 1: Respect; Level 2: Participation and effort; Level 3: Personal Autonomy; Level 4: Help others.

**Table 2 children-08-00573-t002:** Frequencies of model hybridization teaching strategies.

	Teachers (EG) (*N* = 2)		Teachers (CG) (*N* = 2)		U of Mann Whitney
	M	SD	M	SD	*p*-Value
Example of respect	98.33	4.08	100	0.00	0.280
Sets expectations	94.40	6.19	87.23	9.63	0.103
Gives opportunities for success	73.07	9.98	44.19	14.82	0.008 **
Encourages social interaction	76.63	7.41	50.43	16.44	0.006 *
Assigns tasks	21.52	5.60	30.61	17.73	0.517
Leadership	16.75	11.81	17.06	10.23	0.942
Granting of choice and voice	62.32	8.98	15.39	8.17	0.003 **
Role in Evaluation	12.82	4.69	0.00	0.00	0.001 **
Transfer	11.15	7.07	0.00	0.00	0.004 **
Modified sports game	39.93	4.24	41.53	6.43	0.505
Tactical awareness	26.88	10.30	0.00	0.00	0.001 **
Skill execution	23.42	6.98	0.00	0.00	0.001 **

NOTE: * *p* < 0.05; ** *p* < 0.01; EG = Experimental group; CG = Control group; TPSR = Model of Personal and Social Responsibility; TGfU = Teaching Games for Understanding; M = Mean; SD = Standard deviation.

**Table 3 children-08-00573-t003:** Multivariate analysis of the intervention (MANOVA).

		Pretest		Posttest		Difference between Pre and Post	Intergroups Difference of Means
	Group	Mean	SD	Mean	SD	*p*-Value	*p*-Value
AutonomousM	Control	5.636	0.762	5.711	0.804	0.504	0.001 **
	Experimental	5.824	0.919	6.304	0.575	0.000 **	
	*p*-value + η2	0.270	0.013	0.000 **	0.149		
ControlledM	Control	3.991	1.297	4.064	1.335	0.672	0.211
	Experimental	3.980	1.300	4.230	1.429	0.194	
	*p*-valor + η2	0.967	0.000	0.551	0.004		
Demotivation	Control	1.796	0.968	1.623	0.774	0.293	0.011 *
	Experimental	1.994	1.452	1.534	0.852	0.013 *	
	*p*-value + η2	0.417	0.007	0.589	0.003		
SDI	Control	9.350	3.545	9.709	3.526	0.535	0.012 *
	Experimental	9.639	4.945	11.486	3.179	0.005 **	
	*p*-value + η2	0.736	0.001	0.011 *	0.065		
Competences	Control	4.532	1.079	4.323	1.271	0.210	0.887
	Experimental	4.579	1.186	4.824	1.272	0.190	
	*p*-value + η2	0.835	0.000	0.054	0.038		
Relationship	Control	5.296	1.055	5.646	1.028	0.035 *	0.001 **
	Experimental	5.159	0.937	5.693	1.226	0.004 **	
	*p*-value + η2	0.504	0.005	0.834	0.000		
Autonomy	Control	5.030	1.234	5.151	1.024	0.430	0.021*
	Experimental	5.144	1.058	5.561	1.068	0.017 *	
	*p*-value + η2	0.629	0.002	0.056	0.037		
PMI	Control	4.952	0.842	5.040	0.859	0.476	0.009**
	Experimental	4.961	0.792	5.359	0.921	0.004 **	
	*p*-value + η2	0.980	0.000	0.078	0.032		
Enjoyment	Control	4.367	0.578	4.494	0.506	0.162	0.001 **
	Experimental	4.300	0.692	4.659	0.435	0.001 **	
	*p*-value + η2	0.600	0.003	0.090	0.029		
Boredom	Control	1.609	0.820	1.481	0.666	0.244	0.082
	Experimental	1.420	0.828	1.261	0.523	0.193	
	*p*-value + η2	0.260	0.013	0.076	0.032		
Personal responsibility	Control	5.161	0.741	5.200	0.786	0.724	0.040*
	Experimental	5.120	0.816	5.425	0.566	0.015 *	
	*p*-value + η2	0.795	0.001	0.113	0.026		
Social responsibility	Control	5.218	0.667	5.262	0.600	0.610	0.006**
	Experimental	5.081	0.569	5.403	0.598	0.001 **	
	*p*-value + η2	0.273	0.012	0.249	0.014		
IPA	Control	4.029	0.706	4.036	0.826	0.961	0.003 **
	Experimental	4.000	0.646	4.341	0.506	0.000 **	
	*p*-value + η2	0.833	0.000	0.029 *	0.048		

Note: * *p* < 0.05; ** *p* < 0.01; AutonomousM = Autonomous motivation; ControlledM = Controlled motivation; SDI= Self-Determination Index; PMI = psychological mediators index; IPA = intention to be physically active; η2 = Effect size (Cohen’s D); Dif = Difference of means.

## Data Availability

https://osf.io/7h3sm/?view_only=efba01fb288e40b9be76da1ddb5098df, accessed on 18 March 2021.

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
