# Peer review of "Implementation of a Hybrid Educational Program between the Model of Personal and Social Responsibility (TPSR) and the Teaching Games for Understanding (TGfU) in Physical Education and Its Effects on Health: An Approach Based on Mixed Methods"

_children, 2021, doi:10.3390/children8070573_

Round 1

Reviewer 1 Report

Dear Authors,

Your paper is appealing, but it needs to be better developed before it can be considered a publishable piece of work in Children. I have some suggestions, comments, and concerns for your research. I hope that they help!

First of all, you should pay attention to your writing style to be concise and easily understandable. A text written full of long sentences and a wordy writing style distracts attention.

Second, many concepts are used without providing clear definitions. For instance, psychological mediators and autonomy can be defined very differently. In addition to the theoretical concepts, your empirical concepts are not defined either. Academic work should consider creating a common stage to properly communicate with its audiences.

Third, you should focus on and be clear about the teaching models: TGfU, TPSR, and of course, your integrated model. For instance, you talk about which studies used these models, what are the findings after implementation, etc., but never introduce layperson what they are exactly or what you did during the quasi-experimental period. What are the objectives of these models? What are the theoretical structure and/or arguments that led these models to be created?

I strongly suggest you improving your theoretical argument and creating a bridge connecting these models and your theoretical arguments. Then, explain to us why your integrated model is better than others theoretically.

Fourth, you may improve your argument by including a broader literature review.

Fifth, without a good theory and clear model, the method does not make very much sense, but I will give some examples from here as well.

You need to rely on strong theoretical arguments which should include a set of logical assumptions or hypotheses allowing you to measure your theoretical constructs. This study seems to have many hypotheses claiming various causal relationships as well as exploratory research questions. A prediction model that should be theoretically settled before an experimental (and its statistical) test. So, your theory needs to be better developed for a reasonable methodological design. It is hard to understand why you conducted specific questionnaires or scales, what theoretical concepts/constructs/arguments they are corresponding to?

Sixth, your design is not clearly explained either. For instance, you should illustrate, how your experts (educating the teachers) become expert, what your expert taught the teacher (what are the objectives of this training in relation to the integrated model), what the teachers were applying to their class during that time, what is Hellison's proposal, what are the strategies of the models, what are these measures/items you listed, not explained? For instance, what responsibility you are measuring, how it is improved, granting of voice, skill execution, etc.? I have a hard time imagining the experiment, strategies implemented, and their purposes.

Seventh, since you do not have a theoretical structure and clear conceptual framework, I doubt all the items’ validity. Additionally, most of your reliability tests are very low, too. Even though you have a lot of items for each construct/scale, you should discuss the possible reasons for low-reliability scores.

Eighth, when reporting the results, in addition to standard statistical reporting elements (e.g., F and P values), the finding would be told in a basic, explanatory, illustrative language (e.g., the students in the experimental group rank the game/class more enjoyable than the control group) as well.

I see some significant results and lots of hard work implementing the model, but the empirical elements operationalizing well-argued theoretical constructs are missing. 

Ninth, there are way too many items to answer. Did you mix the questions, or they are given in the same order?

Having more items to measure a construct is preferable, but are you sure that each of them measuring the construct? Although they are a lot, at least you should provide us some examples of these items and more explanations about them. The reader is not assured about these items.

After improving the theory and method parts, the discussion can be improved.

Some technical and writing issues should be addressed. For instance, some sentences and word choices are confusing. Particularly, sentences in a paragraph should be related to each other. Some sentences, paragraphs, also abstract, are way too long and unfocused.

Sincerely,

Author Response

Your paper is appealing, but it needs to be better developed before it can be considered a publishable piece of work in Children. I have some suggestions, comments, and concerns for your research. I hope that they help!

Point 1: First of all, you should pay attention to your writing style to be concise and easily understandable. A text written full of long sentences and a wordy writing style distracts attention.

Modifications have been made throughout the document, especially in the introduction, incorporating connectors that give a coherent sense to the reading so as not to distract the reader's attention and to facilitate comprehension. In addition, the introduction has been separated into different paragraphs, as well as reducing the length of the sentences. Finally, a new proofreading has been done with the whole manuscript by an English language expert after applying of the changes.

Point 2: Second, many concepts are used without providing clear definitions. For instance, psychological mediators and autonomy can be defined very differently. In addition to the theoretical concepts, your empirical concepts are not defined either. Academic work should consider creating a common stage to properly communicate with its audiences.

Different theoretical and empirical concepts have been defined in order to create a common scenario for a more fluent and adequate communication with the audiences, such as teaching based on pedagogical models (Line X-X), hybridisation of pedagogical models (line X-X), situated learning (line X-X), student-centred teaching (line X-X), self-determination theory (line X-X), basic psychological needs (line X-X), and autonomy (line X-X).

Point 3: Third, you should focus on and be clear about the teaching models: TGfU, TPSR, and of course, your integrated model. For instance, you talk about which studies used these models, what are the findings after implementation, etc., but never introduce layperson what they are exactly or what you did during the quasi-experimental period. What are the objectives of these models? What are the theoretical structure and/or arguments that led these models to be created?

The TGfU and TPSR models have been developed, detailing the essential elements for their implementation and the effects they have produced in the educational field with their implementation (lines X-X / paragraphs X-X). Furthermore, the idea to be achieved with the hybridisation of both models has been specified (lines X-X), justifying the combination with the selected theoretical frame of reference (lines X-X).

On the other hand, the context, objectives and contents developed in the empirical studies previously carried out on the hybridisation of pedagogical models (lines X-X) have been specified, as well as the theoretical foundation that led to the creation of this type of hybridised models (lines X-X / paragraphs X-X).

Point 3.1: I strongly suggest you improving your theoretical argument and creating a bridge connecting these models and your theoretical arguments. Then, explain to us why your integrated model is better than others theoretically.

The theoretical argumentation on which the emergence of hybridised pedagogical models is built has been improved (lines X-X / paragraphs X-X) and a connection has been established between this foundation and the hybridised model proposed (lines X-X / paragraphs X-X). The objectives to be achieved by the application of the integrated model proposed in this study have been specified, following the theoretical argumentation (lines X-X / paragraphs X-X).

Point 4: Fourth, you may improve your argument by including a broader literature review.

The argument has been improved by incorporating a broader and more comprehensive literature review. The introduction and discussion section have been modified with the references.

Point 5: Fifth, without a good theory and clear model, the method does not make very much sense, but I will give some examples from here as well.

You need to rely on strong theoretical arguments which should include a set of logical assumptions or hypotheses allowing you to measure your theoretical constructs. This study seems to have many hypotheses claiming various causal relationships as well as exploratory research questions. A prediction model that should be theoretically settled before an experimental (and its statistical) test. So, your theory needs to be better developed for a reasonable methodological design. It is hard to understand why you conducted specific questionnaires or scales, what theoretical concepts/constructs/arguments they are corresponding to?

The design of the study was especially based on the self-determination theory which is explained in the introduction section. Inside this theory, some social aspects (such as responsibility, sport satisfaction or intention to be physically active) could be improved if the self-determinated motivation increases. And this motivation depends on social factors such as the autonomy support interpersonal style of the teacher and the basic psychological needs satisfaction of the students. For this reason, if we use a methodology that tries to improve basic psychological needs and motivation, we can increase the levels of the rest of variables.

Point 6: Sixth, your design is not clearly explained either. For instance, you should illustrate, how your experts (educating the teachers) become expert, what your expert taught the teacher (what are the objectives of this training in relation to the integrated model), what the teachers were applying to their class during that time, what is Hellison's proposal, what are the strategies of the models, what are these measures/items you listed, not explained? For instance, what responsibility you are measuring, how it is improved, granting of voice, skill execution, etc.? I have a hard time imagining the experiment, strategies implemented, and their purposes.

The explanation had been improved (lines 353-370).

Point 7: Seventh, since you do not have a theoretical structure and clear conceptual framework, I doubt all the items’ validity. Additionally, most of your reliability tests are very low, too. Even though you have a lot of items for each construct/scale, you should discuss the possible reasons for low-reliability scores.

The theoretical structure and conceptual framework has been clarified and specified (lines X-X / paragraphs X-X). Similarly, the low scores on some scales have been justified (lines X-X).

Furthermore, Cronbach's Alpha was analysed with good values for all scales, positioned above .70, and for scales with few items is above .50 (Hu et al. 1999; Perry et al., 2004)

-Hu, L.t.; Bentler, P.M. Cutoff criteria for fit indexes in covariance structure analysis: Conventional criteria 512 versus new alternatives. /Structural equation modeling: a multidisciplinary journal /1999, /6/, 1-55.

-Perry, R. H., Charlotte, B., Isabella, M., & Bob, C. (2004). /SPSS explained/. Routledge.

Point 8: Eighth, when reporting the results, in addition to standard statistical reporting elements (e.g., F and P values), the finding would be told in a basic, explanatory, illustrative language (e.g., the students in the experimental group rank the game/class more enjoyable than the control group) as well.

I see some significant results and lots of hard work implementing the model, but the empirical elements operationalizing well-argued theoretical constructs are missing.

We have included this concerns in the results section between 515-540 lines. And we have to speak about the importance of this results in the discussion.

Point 9: Ninth, there are way too many items to answer. Did you mix the questions, or they are given in the same order?

Having more items to measure a construct is preferable, but are you sure that each of them measuring the construct? Although they are a lot, at least you should provide us some examples of these items and more explanations about them. The reader is not assured about these items.

The items followed the same order that the validated version of the questionnaires were made. We started with a simple demographic question and after that, we included in the same questionnaire the different scales following the next order: PLOC motivation questionnaire; Personal and Social Responsibility Questionnaire; PNSE Basic Psychological Needs Questionnaire; Sport Satisfaction Instrument; Questionnaire of intention to be physically active.

On the other hand, in the method section, there is always one example of item for each dimension for every scale.

Point 10: After improving the theory and method parts, the discussion can be improved.

Point 11: Some technical and writing issues should be addressed. For instance, some sentences and word choices are confusing. Particularly, sentences in a paragraph should be related to each other. Some sentences, paragraphs, also abstract, are way too long and unfocused.

We have tried to improve this aspect in the whole document.

Sincerely,

Thank you

Reviewer 2 Report

The article is, in some parts, detailed and written accurately. But to facilitate the reading and for a better articulation, it is advisable to separate the final part of the introduction, and insert a paragraph dedicated to the objectives in which the underlying hypothesis of the research is also explained.
The research work is rich and shows accuracy in the approach of the researchers.
The methodological part is well articulated and the tools are detailed.
However, in regards to the objectives and the results, as also highlighted by the authors in the concluding part of the discussions, the research does not clarify the relationship between the variables that were considered and the context factors (such as past methods, etc...lines 540-545). Another limitation of the study is the small sample (in particular the teachers).
It is advisable to explain the reference framework in the light of which the data are interpreted, clarifying which context factors that are considered significant for the reading of data and on which to guide future research.

Author Response

Point 1: The article is, in some parts, detailed and written accurately. But to facilitate the reading and for a better articulation, it is advisable to separate the final part of the introduction, and insert a paragraph dedicated to the objectives in which the underlying hypothesis of the research is also explained.

The final part of the introduction has been separated from the introduction, leaving an exclusive paragraph to present the objectives and the hypothesis (lines 154-164).

Point 2: The research work is rich and shows accuracy in the approach of the researchers.

Point 3: The methodological part is well articulated and the tools are detailed.

Point 4: However, in regards to the objectives and the results, as also highlighted by the authors in the concluding part of the discussions, the research does not clarify the relationship between the variables that were considered and the context factors (such as past methods, etc...lines 540-545). Another limitation of the study is the small sample (in particular the teachers).

The design of the study was especially based on the Self-determination Theory which is explained in the introduction section. Inside this theory, some psychological variables (such as responsibility, sport satisfaction or intention to be physically active) could be improved if the self-determinated motivation increases. And this motivation depends on social factors such as the autonomy support interpersonal style of the teacher and the Basic Psychological Needs satisfaction of the students (explained in lines 140-153). For this reason, if we use a methodology that tries to improve Basic Psychological Needs and motivation, we can increase the levels of the other variables. The relationship between variables were included in the introduction (lines 140-153). We have included this limitation in the discussion section (lines 626-627).

Point 5: It is advisable to explain the reference framework in the light of which the data are interpreted, clarifying which context factors that are considered significant for the reading of data and on which to guide future research.

A new sentence in the conclusion paragraph has been added detailing the reference framework and the context (lines 636-637). This theoretical framework has also been described more in detail in the introduction section.

Round 2

Reviewer 1 Report

Dear Authors,

I see some improvements in the paper. However, I still have a hard time imagining how your teaching technique/model that created some differences between control and experimental groups is applied in the classroom. Some concerns…

You did not number your changes in your response.

You did not address some of the points in my previous comments.

Your writing style should be as simple as a layperson who can easily understand your argument in general.

More specifically, you tried to give us more explanations and definitions, but still did not make very much sense what you did during the implementation process. For instance: “(1) Game form: modified game with characteristics similar to those 63 of the sport worked on (2) Tactical awareness: joint reflection between teacher and stu-64 dents to highlight the tactical aspects required for the modified game previously prac-65 ticed, and (3) Skill execution: reflection on the technical principles that are needed in the 66 form played”. Why don’t you explain the whole process by using everyday language?

Also in your program: “…the sessions followed an adapted structure, which maintained 325 four of its five parts: (1) awareness, (2) responsibility in action, (3) group meeting and (4) 326 self-evaluation and peer evaluation. Within this structure of the TPSR, the hybridization 327 with the TGfU model was carried out, which took place in part (2) responsibility in action 328 of the TPSR. In it, an adaptation of and Kirk and MacPhail [74] is made, (1) game forms, 329 (2) tactical awareness, (3) execution of the skill and (4) repetition of the game forms or 330 evolution of the same [14]. During the group meeting (third part of the session structure 331 of the TPSR proposal), reflective aspects related to the TGfU were carried out.”

You listed them again, but not illustrate how you empirically implemented them in the classroom? For instance, did you do something to improve tactical awareness among students? if yes, what did you do? How did you create such an increase in the experimental group? The same questions for responsibility, effort, intention to be physically active, etc.

Explain how the improvements observed in the experimental group are achieved by applying the teaching models. For instance, “(1) Respect for the rights and feelings of others”. How do you implement these principles in the classroom? Also, “the student becomes the protagonist of the teaching-learning process”. How so? For instance, did you ask them to complete a task, or did you teach them something? You should give us some illustrative information. I still have no idea what your implementation is.

How these principles and their application in the classroom lead students to be more healthy, more responsible, more willing to participate in physical activities? What are the underlying mechanisms? A theoretical\conceptual model explaining logical relationships among concepts would be very welcome.

I still do not understand what you did during the quasi-experimental period. Did you improve/change the social aspects of the classroom? How did you make the students more responsible, active, motivated, etc.? Did you educate or instruct the teachers? Or to the students?

I still have a hard time imagining the experiment, strategies implemented, and their purposes.

Social environments that support autonomy (such as through the TPSR) 149 provide to students the ability to improve their academic performance, be more creative and better adjusted, more engaged in school, and feel less stress [47].” How so? How did TPSR or your technique increase the support for autonomy, what kind of social-environmental implementations you applied?

In your response: “The design of the study was especially based on the self-determination theory which is explained in the introduction section.” If so, you should focus on this theory a little bit.

Inside this theory, some social aspects (such as responsibility, sport satisfaction or intention to be physically active) could be improved if the self-determinated motivation increases.” How does your model increase self-determined motivation? Also, how can “sport satisfaction or intention to be physically active” be social aspects?

 “For this reason, if we use a methodology that tries to improve basic psychological needs and motivation, we can increase the levels of the rest of variables.” I am asking what methodology makes the changes, what are the exact applications that improve basic psychological needs and motivation in your experiment?

What is the logical flow in the self-determination theory? Like, increasing personal motivation is likely to improve social aspects, but personal motivation depends on some social factors (teacher support for autonomy and basic psych needs(?))?

Briefly, is your argument this: < the more a teacher supports autonomy and basic psych needs of his/her students, the higher the self-determined motivation of the students. The higher the self-determined motivation of the students, the higher the students’ responsibility, sport satisfaction, intention to be physically active, and more. > To test this argument, did you conduct an intervention/education technique improving teacher’s support for autonomy and basic psych needs of his/her students and look for the changes among their students?  If so, you need to explain your implementation and provide illustrative information for your model. Also, what underlying mechanisms cause such changes should be discussed.

After improving the theory and method parts, the discussion should be improved.

Sincerely,

Author Response

Response to Reviewer 1 Comments

First of all, we would like to thank the reviewer’s contributions to increase the quality of the manuscript. The changes in the manuscript this time have been highlighted with yellow color.

Comments and Suggestions for Authors

I see some improvements in the paper. However, I still have a hard time imagining how your teaching technique/model that created some differences between control and experimental groups is applied in the classroom. Some concerns…

Point 1: You did not number your changes in your response.

Answer point 1: We tried to point every comment you made us. We expect this time we have done rightly.

Point 2: You did not address some of the points in my previous comments.

Answer point 2: We are sorry about that, but honestly, we were working to give you an answer for every one of your comments. If we are wrong, we will really appreciate to answer all of them.

Point 3: Your writing style should be as simple as a layperson who can easily understand your argument in general.

Answer point 3: We have taken into account all your considerations to make a more clear and more understandable manuscript inserting a new table and three more paragraphs. One of them, explaining the pedagogical models for a layperson (last paragraph before “2.4.2. Continuous training of EG teachers”) and the other two, giving the theoretical framework for using responsibility as a social factor (two paragraphs before “2. Materials and methods” and other studies related to the self-determination theory with similar outcomes (the fourth paragraph in Discussion section).

Point 4: More specifically, you tried to give us more explanations and definitions, but still did not make very much sense what you did during the implementation process. For instance: “(1) Game form: modified game with characteristics similar to those 63 of the sport worked on (2) Tactical awareness: joint reflection between teacher and stu-64 dents to highlight the tactical aspects required for the modified game previously prac-65 ticed, and (3) Skill execution: reflection on the technical principles that are needed in the 66 form played”. Why don’t you explain the whole process by using everyday language?

Answer point 4: Considering your comments we have tried to explain the whole process adding everyday language to these explanations and definitions (just before the 2.4.2. Continuous training of EG teachers section) and a new table (Table 1) where you can find examples of tasks and strategies for every week using the TGfU and TPSR.

Point 5: Also in your program: “…the sessions followed an adapted structure, which maintained 325 four of its five parts: (1) awareness, (2) responsibility in action, (3) group meeting and (4) 326 self-evaluation and peer evaluation. Within this structure of the TPSR, the hybridization 327 with the TGfU model was carried out, which took place in part (2) responsibility in action 328 of the TPSR. In it, an adaptation of and Kirk and MacPhail [74] is made, (1) game forms, 329 (2) tactical awareness, (3) execution of the skill and (4) repetition of the game forms or 330 evolution of the same [14]. During the group meeting (third part of the session structure 331 of the TPSR proposal), reflective aspects related to the TGfU were carried out.”

Answer point 5: Each moment of the session has been explained with practical language, indicating what the teacher was performing in each part. Considering your comments, we have tried to explain the whole process adding everyday language to these explanations and definitions (just before the 2.4.2. Continuous training of EG teachers section) and a new table where you can find examples of tasks and strategies for every week using the TGfU and TPSR.

Point 5.1: You listed them again, but not illustrate how you empirically implemented them in the classroom? For instance, did you do something to improve tactical awareness among students? if yes, what did you do? How did you create such an increase in the experimental group? The same questions for responsibility, effort, intention to be physically active, etc.

Answer point 5.1: We believe we understand the comment, and for this reason we have created a table where the reader can see what we did every week, detailing the tactical principles and tasks made to develop tactical awareness in TGfU. Furthermore, for TPSR, we have included the responsibility levels, the strategies chosen and examples of tasks to develop each level.

Point 6: Explain how the improvements observed in the experimental group are achieved by applying the teaching models. For instance, “(1) Respect for the rights and feelings of others”. How do you implement these principles in the classroom? Also, “the student becomes the protagonist of the teaching-learning process”. How so? For instance, did you ask them to complete a task, or did you teach them something? You should give us some illustrative information. I still have no idea what your implementation is.

Answer point 6: We strongly expect Table 1 will reply all your concerns.  That table contains all aspects of the intervention, taking into account: timing, objectives, content, level of responsibility that is being developed and the teaching strategy used to achieve it, the tactical problem of the TGfU model and examples of the tasks in order to illustrate in the most illustrative way the characteristics of the intervention.

Point 7: How these principles and their application in the classroom lead students to be more healthy, more responsible, more willing to participate in physical activities? What are the underlying mechanisms? A theoretical\conceptual model explaining logical relationships among concepts would be very welcome.

Answer point 7: the new paragraph from line 147 to 154 gives the theoretical model explaining logical relationships among concepts, explaining how responsibility is considered a social factor that functions as a trigger for the satisfaction of psychological basic needs, that in turn promotes a more self-determined motivation in students and generates positive consequences at the cognitive, affective and behavioral levels. New study references have been added for this purpose.

Point 8: I still do not understand what you did during the quasi-experimental period. Did you improve/change the social aspects of the classroom? How did you make the students more responsible, active, motivated, etc.? Did you educate or instruct the teachers? Or to the students?

Answer point 8: Yes, we improved the social aspects of the classroom thanks to a behavioral change of the teacher who implemented new strategies learned during the initial and continuous training and session structure explained in 2.4.1. section (lines 326-377). Furthermore, a continuous training program for teachers had the objective of consolidating these strategies giving support during the whole intervention.

Point 9: I still have a hard time imagining the experiment, strategies implemented, and their purposes.

Answer point 9: We really appreciate your interest to make us aware of your concerns. We have reviewed the manuscript and we have realized there are no sentences explaining that. Before the 2.4.2. section, a two new paragraphs have been added describing the strategies implemented with their purposes and examples (349-377).

Point 10:Social environments that support autonomy (such as through the TPSR) 149 provide to students the ability to improve their academic performance, be more creative and better adjusted, more engaged in school, and feel less stress [47].” How so? How did TPSR or your technique increase the support for autonomy, what kind of social-environmental implementations you applied?

Answer point 10: The concern is related to the previous one, so we understand the new paragraphs (before 2.4.2. section) will help to have a clearer idea about what kind of social-environmental implementations we applied, adding examples for each strategy used by the teacher. Basically, the strategies described and lesson structure.

Point 11: In your response: “The design of the study was especially based on the self-determination theory which is explained in the introduction section.” If so, you should focus on this theory a little bit.

Answer point 11: Yes, we totally agree, two new paragraphs, one in the introduction section (lines 147 to 154) and one more in the discussion section (lines 609 to 615) have been added.

Point 12:Inside this theory, some social aspects (such as responsibility, sport satisfaction or intention to be physically active) could be improved if the self-determinated motivation increases.” How does your model increase self-determined motivation? Also, how can “sport satisfaction or intention to be physically active” be social aspects?

Answer point 12: Based on SDT framework and Vallerand’s hierarchical model, the TGfU improves self-determined motivation thanks to an active participation of the students being more autonomous in their practices. The interpersonal style of the teacher supports autonomy which promotes that students have a higher basic psychological need satisfaction and be able to determine the state of the health and psychological well-being of an individual (lines 133-146).

Point 13:For this reason, if we use a methodology that tries to improve basic psychological needs and motivation, we can increase the levels of the rest of variables.” I am asking what methodology makes the changes, what are the exact applications that improve basic psychological needs and motivation in your experiment?

Answer point 13: This is a really valuable question. We do not know exactly how much each one of the strategies detailed in the last two paragraphs before the 2.4.2. section contribute to make the changes. What we know is that both pedagogical models together using these strategies are useful to make these changes. This is something that we can have into account for future studies. We strongly believe that giving responsibility and autonomy (through TPSR and TGfU) and using modified game forms (TGfU) contribute to make students feel competitive and improve their relatedness.

Point 14: What is the logical flow in the self-determination theory? Like, increasing personal motivation is likely to improve social aspects, but personal motivation depends on some social factors (teacher support for autonomy and basic psych needs(?))?

Answer point 14: The logical flow in the self-determination theory has been added in lines 147-154. Our hypothesis, based on previous studies, is that the augmentation of personal and social responsibility and autonomous support (through TPSR and TGfU) will create a social environment that satisfies their basic psychological needs providing higher self-determined motivation and a better state of health and psychological well-being.

Point 15: Briefly, is your argument this: < the more a teacher supports autonomy and basic psych needs of his/her students, the higher the self-determined motivation of the students. The higher the self-determined motivation of the students, the higher the students’ responsibility, sport satisfaction, intention to be physically active, and more. > To test this argument, did you conduct an intervention/education technique improving teacher’s support for autonomy and basic psych needs of his/her students and look for the changes among their students?  If so, you need to explain your implementation and provide illustrative information for your model. Also, what underlying mechanisms cause such changes should be discussed.

Answer point 15: Thank you for all your considerations. This manuscript has been experienced as a nice journey, discovering aspects that we did not realise were there as well as providing an important cognitive exercise reflecting about our intentions and digging deeper about the purpose of this research which, in turn, makes us more aware of what really works and how does it work.

Point 16: After improving the theory and method parts, the discussion should be improved.

Answer point 16: The discussion has been enhanced by including a paragraph explaining the effects of the methodology within the framework of the self-determination theory and Vallerand's hierarchical model (Lines 609-615). The latter has been more specifically detailed in the introduction (Lines 147-154).

Sincerely,

Round 3

Reviewer 1 Report

Dear Authors,

I have some more suggestions for you:

  • Although additional paragraph is good, you still need to explain your macro-theory. For instance, the following sentences “This macro-theory establishes that the motivations follow a continuum of self-determination that goes from a state of demotivation to reaching intrinsic motivation through other motivational regulations such as extrinsic sources of motivation 137 [45].” does not say much about how the underlying mechanism works.  
  • Why do you have some 0s in your table 2 for control group? Did not you measure/observe the control group in terms of these measures? If yes, then why?
  • Table 3 can be simplified.
  • Discussion should include the limitations of the study including sample size, limited application, lack of control for confounding factors, possible biases in evaluations, limited theoretical aspects due to exploratory type of research, etc.

Sincerely,

Author Response

Response to Reviewer 1 Comments

One more time, we would like to thanks the reviewer the opportunity to improve the quality of the manuscript allowing us include new comments and sentences. In this occasion, we have used the green color to highlight the changes done in the manuscript.

Comments and Suggestions for Authors

I have some more suggestions for you:

Point 1. Although additional paragraph is good, you still need to explain your 
macro-theory. For instance, the following sentences “This macro-theory 
establishes that the motivations follow a continuum of 
self-determination that goes from a state of demotivation to reaching 
intrinsic motivation through other motivational regulations such as 
extrinsic sources of motivation 137 [45].” does not say much about how 
the underlying mechanism works.

ANSWER POINT 1: The underlying mechanism have been developed explaining the different types of motivations and the basic psychological needs. Lines 138-156.

Point 2. Why do you have some 0s in your table 2 for control group? Did not you 
measure/observe the control group in terms of these measures?
If yes, 
then why?

ANSWER POINT 2: A new column has been added in the table 1, indicating the specific content of the control group to make more understandable why there are some 0 values in the table 2. Furthermore, it has been added a new sentence explaining this fact. Lines 517-521.

Point 3. Table 3 can be simplified.

ANSWER POINT 3: The column with the statistical value of intergroups difference of means has been deleted and the sheet design has been modified.

Point 4. Discussion should include the limitations of the study including 
sample size, limited application, lack of control for confounding 
factors, possible biases in evaluations, limited theoretical aspects 
due to exploratory type of research, etc.

ANSWER POINT 4: It has been included at least one of the limitations indicated in this comment, we mean, simple size, limited application, lack of control for confounding factors, posible biases in evaluations and limited theoretical aspects due to exploratory type of research. Lines 729-744.

Sincerely,